# Augmentation of the Benzyl Isothiocyanate-Induced Antiproliferation by NBDHEX in the HCT-116 Human Colorectal Cancer Cell Line

**DOI:** 10.3390/ijms26178145

**Published:** 2025-08-22

**Authors:** Ruitong Sun, Aina Yano, Ayano Satoh, Shintaro Munemasa, Yoshiyuki Murata, Toshiyuki Nakamura, Yoshimasa Nakamura

**Affiliations:** 1Graduate School of Environmental and Life Science, Okayama University, Okayama 700-8530, Japan; ptvn5m6i@s.okayama-u.ac.jp (R.S.); muta@cc.okayama-u.ac.jp (Y.M.);; 2School of Food Science and Technology, Dalian Polytechnic University, Dalian 116034, China; 3Graduate School of Interdisciplinary Science and Engineering in Health Systems, Okayama University, Okayama 700-8530, Japan; ayano113@cc.okayama-u.ac.jp; 4Graduate School of Environmental, Life, Natural Science and Technology, Okayama University, Okayama 700-8530, Japan

**Keywords:** benzyl isothiocyanate, multidrug resistance, glutathione *S*-transferase, NBDHEX, apoptosis, c-Jun N-terminal kinase

## Abstract

Increased drug metabolism and elimination are prominent mechanisms mediating multidrug resistance (MDR) to not only chemotherapy drugs but also anti-cancer natural products, such as benzyl isothiocyanate (BITC). To evaluate the possibility of combined utilization of a certain compound to overcome this resistance, we focused on glutathione *S*-transferase (GST)-dependent metabolism of BITC. The pharmacological treatment of a pi-class GST-selective inhibitor, 6-(7-nitro-2,1,3-benzoxadiazol-4-ylthio)hexanol (NBDHEX), significantly increased BITC-induced toxicity in human colorectal cancer HCT-116 cells. However, NBDHEX unexpectedly increased the level of the BITC–glutathione (GSH) conjugate as well as BITC-modified proteins, suggesting that NBDHEX might increase BITC-modified protein accumulation by inhibiting BITC–GSH excretion instead of inhibiting GST. Furthermore, NBDHEX significantly potentiated BITC-induced apoptosis with the enhanced activation of apoptosis-related pathways, such as c-Jun N-terminal kinase and caspase-3 pathways. These results suggested that combination treatment with NBDHEX may be an effective way to overcome MDR with drug efflux and thus induce the biological activity of BITC at lower doses.

## 1. Introduction

Benzyl isothiocyanate (BITC), an aromatic isothiocyanate (ITC) from green papaya fruits [1], inhibits the cell proliferation of many types of cancerous cells via the signaling pathways related to apoptosis induction and cell cycle arrest [2,3,4]. For example, BITC induces apoptosis through the mitochondrial death pathway with the caspase-9/3 activation [2,3,5,6,7]. BITC also activates the mitogen-activated protein kinases (MAPKs) in various cancer cells [2,8,9,10,11]. Among them, c-Jun N-terminal kinase (JNK) is mainly involved in BITC-induced apoptosis [2,8,9,10]. JNK activated by BITC enhances the phosphorylation of Bcl2, an anti-apoptotic regulator, concomitantly upregulating the expression of the pro-apoptotic Bax protein [2].

Innate and acquired resistance in cancer cells can limit the effectiveness of anticancer drugs, leading to treatment failure and thus disease progression [4,12]. The enhanced activities of drug detoxification and efflux, as well as hyperactivation of cell survival and proliferation pathways, are among the most frequently encountered manners by which cancer cells acquire multidrug resistance (MDR) [12]. There are several enzymes that catalyze phase II drug detoxifying reactions, including glucuronosyltransferases, sulfotransferases, acetyltransferases, and glutathione *S*-transferases (GSTs) [13]. Among them, GSTs play an important role in the detoxification of toxic electrophilic chemicals, including anticancer drugs as well as ITCs, by conjugating the reduced form of glutathione (GSH) with them [14]. Especially, pi-class GSTs including GSTP1 are overexpressed in a variety of cancers, including ovarian, breast, colon, and pancreatic cancers [15], and the preclinical studies using human colon cancer cell lines lacking GSTP1 have shown that GSTP1 is important for the survival and growth of human colon cancer cells [16]. Among the ATP-binding cassette transporters closely linked to the drug efflux, multidrug resistance-associated proteins (MRPs) play important roles in the transport of the GSH-conjugated drug metabolites [17,18]. Therefore, GSTs not only decrease the intracellular concentration of the active drugs, but also synergistically act with MRPs to promote drug efflux, further exacerbating MDR.

Because the MDR mechanism may also reduce the ability of BITC to exhibit antiproliferative effects, the concentration required for BITC to exhibit anticancer activity is relatively higher than that of typical anticancer compounds of natural origin [4]. ITCs, including BITC, undergo conjugation with GSH via GSTs in the intestine and liver to form dithiocarbamate metabolites, which are then sequentially metabolized by the mercapturic acid pathway and eliminated via MRPs [2,19]. Since BITC can be metabolized by GSTs, the intracellular concentration of BITC might be regulated by the GST enzyme activity. The purpose of this study is to explore compounds that improve the MDR mechanism and enable BITC to exert its effect at lower concentrations through combination treatment. We have thus focused on a pi-class GST-selective inhibitor, 6-(7-nitro-2,1,3-benzoxadiazol-4-ylthio)hexanol (NBDHEX), as an agent to improve the anticancer effect of BITC in human colorectal cancer HCT-116 cells. To clarify the role of pi-class GSTs in the mechanism of BITC resistance in human colorectal cancer cells, we investigated the effect of NBDHEX on the intracellular accumulation of BITC metabolites. Finally, we found that the combined treatment with NBDHEX and BITC significantly increased the proportion of apoptotic cells. The present data indicate that NBDHEX might increase the BITC-modified protein accumulation by inhibiting BITC–GSH excretion instead of by inhibiting GST. Thus, combination treatment with NBDHEX may be a promising strategy to overcome MDR via the inhibition of drug efflux and thus induce the biological activity of BITC at lower doses.

## 2. Results

### 2.1. Enhancing Effects of NBDHEX on BITC-Induced Antiproliferation

The HCT-116 cell line is commonly used as a colorectal cancer model for drug efflux with overexpression of GSTP1 [20,21]. We used HCT-116 cells in the present study since they are more resistance to BITC than colorectal cancer cell lines with mutant p53 [4]. NBDHEX, a representative 7-nitro-2,1,3-benzoxadiazole derivative, selectively inhibits the representative pi-class GST, GSTP1 [22]. Initially, the effects of BITC and NBDHEX itself on the viability of HCT-116 cells was evaluated using an MTT assay. Here, 24-hour incubation with BITC significantly reduced cell viability in a dose-dependent manner (Figure 1A). Similarly, NBDHEX alone decreased cell viability in a dose-dependent manner (Figure 1B). We thus examined the effect of NBDHEX on BITC-induced antiproliferation in HCT-116 cells. We used 10 μM BITC and 0.5 μM NBDHEX since they slightly reduced cell viability. In this study, NBDHEX alone reduced cell viability insignificantly by only approximately 4.0 ± 0.7%, whereas BITC alone slightly but significantly inhibited the cell proliferation by 17.3 ± 1.5% (Figure 1C). The combination of NBDHEX with BITC resulted in a more pronounced decrease in cell viability by 38.3 ± 4.0%. This effect was comparable to that induced by 20 μM BITC alone, suggesting that NBDHEX was able to halve the effective concentration of BITC. This result is supported by the comparison of the IC50 values, which decreased from 17.1 μM for BITC alone to 7.25 μM for the combination with NBDHEX (Appendix A).

To investigate whether GSTP1/2 plays a role in reducing the efficacy of BITC in human colorectal cancer cells, an experiment using human liver cancer HepG2 cells, which express GSTP1/2 only below the detection limit (Figure 1D), was conducted. As shown in Figure 1E, NBDHEX also significantly enhanced BITC-induced antiproliferation in HepG2 cells. These results suggest that NBDHEX is effective not only in human colorectal cancer cells but also in human liver cancer cells and that its effects are mediated by mechanisms other than the inhibition of GSTP1/2 isozymes.

### 2.2. Modulatory Effect of NBDHEX on the Intracellular Metabolism of BITC

We investigated the effect of NBDHEX on the total GST activity of the cell lysate of HCT-116 cells. As shown in Figure 2A, NBDHEX significantly but only partly inhibited the total GST activity. BITC did not significantly inhibit it even at a concentration of 10 μM (Figure 2B). We next checked the total BITC levels in the HCT-116 cells pretreated with NBDHEX. The peak level of the intracellular BITC accumulation was observed at 1 h after the BITC treatment followed by a gradual decrease. Pretreatment with NBDHEX for 1 h did not significantly increase the total BITC concentration compared to the cells treated with BITC alone (Figure 2C).

To investigate whether NBDHEX can modulate the intracellular metabolism of BITC, we first investigated whether NBDHEX inhibits the conjugation reaction between BITC and GSH. Contrary to expectation, NBDHEX did not suppress the accumulation of BITC–GSH but rather promoted it (Figure 2D), suggesting that NBDHEX could not inhibit BITC–GSH conjugation. We then examined the effect of NBDHEX on the intracellular GSH level because the intracellular GSH level is reduced by the BITC conjugation [23]. Treatment with BITC alone significantly reduced the GSH level, whereas the pretreatment with NBDHEX counteracted this effect (Figure 2E). Furthermore, the level of the BITC-conjugated proteins, detected using the western blotting method, was also increased by the NBDHEX pretreatment (Figure 2F). These results strongly suggested that NBDHEX might alter BITC metabolism, which is to say, it might inhibit the efflux of the BITC–GSH conjugate and thus increases the intracellular concentration of BITC-modified proteins, which are generated by thiol exchange reactions or by free BITC produced by deconjugation of the BITC–GSH conjugate [2].

### 2.3. Enhancing Effects of NBDHEX on BITC-Induced Apoptosis

Since BITC induces apoptosis through the signaling pathway related to JNK among the MAPKs [8,9,10], we examined the effects of NBDHEX on the phosphorylation of JNK and its substrate c-jun induced by 5 μM BITC, which is enough to significantly increase the phosphorylation of JNK as well as c-jun. Western blot analysis revealed that NBDHEX significantly enhanced the BITC-induced phosphorylation of both JNK and c-jun without affecting their total protein levels, whereas NBDHEX alone did not significantly change them (Figure 3A). We next examined the effect of NBDHEX on the BITC-induced caspase-3 activation, a key event involved in the BITC-triggered apoptosis [5,6,7]. As shown in Figure 3B, NBDHEX pretreatment significantly increased the protein level of the cleaved caspase-3 compared to BITC alone, whereas the native form of caspase-3 was not altered by each treatment. In addition to the western blotting analysis, NBDHEX potentiated the BITC-enhanced caspase-3 enzyme activity, while NBDHEX alone did not affect it (Figure 3C). We further investigated whether NBDHEX enhances BITC-induced apoptosis in HCT-116 cells using the Annexin V-FITC staining combined with fluorescence microscopy. As shown in Figure 3D, treatment with NBDHEX (0.5 μM) or BITC (10 μM) alone, slightly but not significantly, increased the proportion of the Annexin V-positive cells. On the other hand, the combined treatment of NBDHEX with BITC significantly increased it compared to each compound alone. These results suggest that NBDHEX enhances the antiproliferative effect of BITC by promoting the induction of apoptosis.

## 3. Discussion

In the present study, NBDHEX was identified as a potentiator of BITC-induced antiproliferation not only in human colorectal cancer HCT-116 cells overexpressing GSTP1 [4] but also in human liver cancer HepG2 cells with no GSTP1/2 expression (Figure 1). The present results also suggest the major role of apoptosis induction and the related signaling pathways in the NBDHEX-enhanced antiproliferative effects (Figure 3). Our group has identified several compounds as improving agents of BITC-induced anticancer effects, such as the MRP inhibitor MK-571 [24], the cholesterol-depleting agent methyl-β-cyclodextrin [25], and the phosphatidylinositide 3-kinase inhibitor LY294002 [26]. The concentration of NBDHEX required for the enhancing effect (0.5 μM, Figure 1C,E and Figure 3D) was much lower than that of these compounds (10~2500 μM), suggesting that NBDHEX is quite promising as a potentiator of not only the anticancer effect but also other biological activities of BITC. NBDHEX has been recognized as an interesting anticancer compound in several cancer models, either alone or in combination with anticancer drugs such as cisplatin and doxorubicin [22,27]. This study adds BITC to the list of compounds whose anticancer activity is enhanced when combined with NBDHEX. Notably, even though NBDHEX itself is toxic, when utilized at one-twentieth the amount of BITC, it can double the anticancer effect of BITC.

We also confirmed the very weak inhibitory effect of NBDHEX on the total GST activity of the cell lysate from the GSTP1-overexpressed HCT-116 cells (Figure 2A). The pretreatment with NBDHEX did not significantly increase the total intracellular BITC level until 3 h after BITC treatment (Figure 2C). However, it increased the intracellular accumulation of BITC–GSH (Figure 2D) and impaired the glutathione depletion by BITC (Figure 2E). These results indicate that NBDHEX might inhibit the efflux of BITC–GSH but not the GSH conjugation of BITC by GSTs. NBDHEX is a competitive inhibitor with a good specificity for the GSH-binding site (G-site) of GST, but rather acts as a suicide substrate-type inhibitor by conjugating with GSH and forming a stable sigma complex on the hydrophobic site (H-site) of GST [22,28]. The H moiety can interact with a variety of hydrophobic toxic compounds, including BITC, which NBDHEX can interfere with [28]. However, this information is inconsistent with the findings in the present study that NBDHEX increased the intracellular amount of the BITC–GSH conjugate. Furthermore, we observed that the level of BITC-modified proteins was significantly increased by pretreatment with NBDHEX (Figure 2F). Taken together, these findings suggest that the accumulated BITC–GSH might contribute to the accumulation of the BITC-modified proteins (Figure 4), which are generated by thiol exchange reactions or by free BITC produced by the deconjugation of BITC–GSH [2]. This implication also supports the finding that NBDHEX restored the intracellular GSH levels depleted by BITC. The unexpected metabolic modification by NBDHEX contributes, at least in part, to the enhancement of the antiproliferation, which is supported by a previous report showing that the BITC-modified protein levels are closely related to the inhibition of cell proliferation by BITC [29,30]. Since NBDHEX has been reported not to be a substrate for p-glycoprotein or MRP1 [31], it is unlikely to competitively inhibit the efflux activity of BITC–GSH. In any case, the mechanism by which NBDHEX increases the accumulation of BITC–GSH remains to be investigated.

Induction of apoptosis is one of the key mechanisms for the anticancer action of BITC [2,3]. The current study demonstrated that NBDHEX significantly enhanced the BITC-induced phosphorylation of JNK and its substrate c-jun in HCT-116 cells (Figure 3A). BITC can activate all the MAPK pathways [8,9,10,11] that mediate intracellular signaling associated with pro-apoptotic and/or anti-apoptotic phenomena [32]. Among the MAPKs, the JNK pathway might be more important than the extracellular signal-regulated kinase and p38 MAPK pathways in the mechanisms underlying BITC-induced apoptosis [2]. This speculation was supported by the findings that an ERK pathway inhibitor showed no significant effect on the BITC-induced antiproliferation in HCT-116 cells [26] and that the p38 MAPK pathway is involved mainly in the BITC-induced cell cycle arrest [2]. Alternatively, GSTP1 negatively regulates the JNK phosphorylation of c-jun and thus downstream signaling [33]. NBDHEX binds to GSTP1, resulting in not only the inhibition of the GST enzyme activity but also counteraction of the JNK inhibition [34]. NBDHEX was also reported to enhance JNK-dependent signaling [35,36]. In this study, NBDHEX increased the BITC-induced activation of the JNK pathway at a concentration that did not activate it by a single treatment (Figure 3A). However, the possibility that NBDHEX enhances the JNK pathway by interfering with the JNK inhibition by GSTP1 could not be completely excluded, because BITC also contributes to the mechanism that inhibits the JNK activation by increasing the GSTP1 enzyme activity and protein expression [35].

In addition, both the proteolytic activation and enzyme activity of caspase-3, an executioner protease for apoptosis induction, were significantly enhanced by the combination of NBDHEX with BITC (Figure 3B,C). Finally, BITC-induced apoptosis was also potentiated by the combination of NBDHEX with BITC, as evaluated by the proportion of the annexin V-positive cells correlated with the caspase-3-dependent increases in cell surface phosphatidylserine (Figure 3D). Taken together, these data suggest the implication that NBDHEX enhances the BITC-induced apoptosis, possibly through enhancement of its intracellular accumulation and/or counteraction of the JNK inhibition by GSTP1. Further studies are warranted to clarify the involvement of NBDHEX in enhancing the BITC action because NBDHEX may also promote the accumulation of the endogenous substrates of GSTP1/2.

In conclusion, NBDHEX could enhance the anticancer effects of BITC through mechanisms other than those commonly expected. NBDHEX at one-twentieth the amount of BITC was able to double the anticancer effect of BITC. Notably, NBDHEX did not inhibit the BITC–GSH conjugation but increased in the amount of the BITC-modified proteins. The current study provides the possibility that the combination of BITC with NBDHEX overcomes the MDR with the drug efflux and thus increases the biological activity of BITC at lower doses. Although NBDHEX has been identified as the pi-class GST-selective inhibitor, one of the significant obstacles that GST inhibitors encounter in clinical trials is their insufficient specificity [37,38]. Although it is likely that GST inhibitors, including NBDHEX, have multiple off-target effects, it cannot be ruled out that some of these effects may lead to beneficial outcomes, as in the present study. Therefore, elucidating these effects is important for future research. On the other hand, the use of the cultured human cancer cell models with HCT-116 or HepG2 cells has considerable limitations. The most important disadvantage is that this model does not reflect the in vivo situation. Next, this model does not consider the possibility that BITC is metabolized not only in the liver but also in the upper gastrointestinal tract. Another limitation of this study is that the experiments attempting to describe the mechanism were conducted using only one cell line, which is not considered to be highly reliable. Ideally, at least two cell lines expressing GST, including normal cells, should be included to prove our implication, which is also a future study. In addition, since BITC itself has toxic effects at relatively higher concentrations, such as the necrosis induction [2], the possibility that these effects are observed at very low concentrations in combination with NBDHEX has not been excluded. Furthermore, the effective concentration of BITC for the antiproliferative effect (~5 μM) might still be far above the physiological concentration [2], even though the combination of BITC with NBDHEX enhanced its activity. Therefore, further investigations are warranted to elucidate the efficacy of the combination of NBDHEX and BITC in in vivo cancer transplantation models. Nevertheless, since NBDHEX has the potential to enhance the BITC-induced biological activities other than anticancer effects through the off-target effects, it is also of interest to study the various biological activities of BITC in cellular models.

## 4. Materials and Methods

### 4.1. Materials

BITC was purchased from LKT Laboratories (St. Paul, MN, USA). NBDHEX was obtained from MedChemExpress (Monmouth Junction, NJ, USA). Antibodies against actin and peroxidase affinipure goat anti-rabbit and anti-mouse IgG (H+L, 115-035-003) were purchased from Jackson ImmunoResearch (West Grove, PA, USA). Antibodies against the phosphorylated-JNK (T183/Y185, #9251), JNK (#9252), caspase-3 (#9662), and cleaved caspase-3 (#9661) were obtained from Cell Signaling Technology (Danvers, MA, USA). Antibodies against the GSTP1/2 (sc-134469), phospho-c-jun (sc-134469), and c-jun (sc-74543) were obtained from Santa Cruz Biotechnology (Santa Cruz, CA, USA). The antibody against BITC-modified proteins was produced as previously reported [39]. Fetal bovine serum (FBS) was obtained from Nichirei Corporation (Tokyo, Japan). Dulbecco’s modified Eagle’s medium (DMEM, high glucose) was purchased from GIBCO/Thermo Fisher Scientific (Waltham, MA, USA). The Bio-Rad Protein Assay was purchased from Bio-Rad Laboratories (Hercules, CA, USA). All other chemicals were obtained from Wako Pure Chemicals Industries (Osaka, Japan) or Nacalai Tesque (Kyoto, Japan).

### 4.2. Cell Culture, Treatment, and Cell Viability Determination

HCT-116 cells and HepG2 cells derived from the American Type Culture Collection were maintained as previously reported [24]. DMSO was used to dissolve and dilute BITC and NBDHEX to the different concentrations. In the treatment experiments, the cells were treated with complete medium containing each reagent or solvent (final 0.1%, *v*/*v*). After HCT-116 cells (5 × 10^4^ per well in a 96-well plate) were preincubated overnight, they were then pretreated with NBDHEX (0.5 μM) for 1 h. After the 24-hour treatment with BITC, the MTT assay was performed in accordance with the previous report [26].

### 4.3. Glutathione S-Transferase (GST) Activity Assay

The total activity of GST was evaluated using a 1-chloro-2,4-dinitrobenzene (CDNB) assay according to the previous report [40]. Protein concentration in the cell lysates was determined using the Bio-Rad Protein Assay. The cell lysate (100 μg protein in 100 mM phosphate buffer, pH 6.5) was mixed with NBDHEX or BITC at the indicated concentrations. Then, 1 mM GSH and 1 mM CDNB (substrate) were added to the cell lysate to start this assay. The CDNB–GSH conjugate was measured by the change in absorbance at 340 nm over 5 min. One unit was defined as the amount of enzyme activity that catalyzed 1 mmol CDNB to its GSH conjugate per minute.

### 4.4. Measurement of Intracellular BITC Accumulation

The intracellular BITC level was quantified using the cyclocondensation assay [24]. The cells (5 × 10^6^) were preincubated overnight in a 60 mm plate and then pretreated with NBDHEX (0.5 μM) for 1 h. After incubation with BITC (20 μM) for the indicated periods, the cell lysates were incubated with 1,2-benzenedithiol at 65 °C for 2 h. The supernatants containing the reaction product, 1,3-benzodithiole-2-thione, were analyzed by reverse-phase high-performance liquid chromatography with ultraviolet detection (HPLC-UV) at 365 nm (Waters ACQUITY UPLC Hclass, Waters Corporation, Milford, MA, USA).

### 4.5. Measurement of Intracellular BITC–GSH Accumulation

The cells were pretreated with NBDHEX for 1 h, followed by the BITC treatment for 1 h, and then collected using a cell scrap (AGC TECHNO GLASS, Shizuoka, Japan). Protein concentration in the cell lysates was determined by the Bio-Rad Protein Assay. After deproteinization of the cell lysates with normalized protein concentration using methanol, the level of BITC–GSH was determined using a reverse-phase HPLC system connected to a triple quadrupole mass spectrometry device (Xevo TQD, Waters Corporation, Milford, MA, USA) using multiple-reaction monitoring with 455.0/306.0 [M+H]^+^ as described previously [1].

### 4.6. Measurement of Intracellular GSH Level

After the overnight preincubation of the cells (2 × 10^6^) in a 60 mm plate, cells were pretreated with NBDHEX (0.5 μM) for 1 h, followed by BITC for 1 h. The total glutathione content was evaluated using glutathione reductase (GR) with 5,5′-dithiobis(2-nitrobenzoic acid) (DTNB) [41]. Fifty microliters of the cell lysate were mixed with 100 μL of reaction solution added in a 24-well plate. Reaction solution contained 2.8 mL of 1 mM DTNB, 3.75 mL of 1 mM NADPH, 5.85 mL of 50 mM KH_2_PO_4_ buffer with 5 mM EDTA (pH 8.0), and 4 units of GR. The change in absorbance was recorded at 415 nm using a microplate reader (SH-9000Lab, CORONA ELECTRIC, Hitachi-Naka, Japan).

### 4.7. Western Blot Analysis

Western blot analysis was used for the evaluation of the protein levels of the GSTP1/2, BITC-modified proteins, JNK, c-jun, and caspase-3 as previously reported [24,35]. After the overnight preincubation of the cells (2 × 10^6^) in a 60 mm plate, cells were pretreated with NBDHEX (0.5 μM) for 1 h, followed by treatment with BITC (5 μM) for 1 h (BITC-modified proteins and JNK/c-jun) or 24 h (caspase-3). LAS-500 (Cytiva, Tokyo, Japan) was utilized to obtain the captured images.

### 4.8. Apoptosis Assay

After the overnight preincubation of HCT-116 cells (2 × 10^5^) in a 12-well plate, cells were pretreated with NBDHEX (0.5 μM) for 1 h. After the 24-h treatment with BITC, the percentage of cell population exhibiting apoptotic characteristics was estimated using a commercially available apoptosis-detecting kit (Annexin V-FITC Apoptosis Detection Kit, Nacalai Tesque Inc., Kyoto, Japan) [24]. A fluorescence microscope was used to analyze the stained cells (Biozero BZ-8000, KEYENCE, Osaka, Japan) with an OP-87763 BZ filter (excitation wavelength 470/40 nm, absorption wavelength 535/50 nm, and dichroic mirror wavelength 495 nm, KEYENCE, Osaka, Japan). In each experiment, more than four images were obtained from three independent plates. The fluorescence images from at least 150 cells were analyzed.

### 4.9. Statistical Analysis

All the values are shown as the mean ± SD obtained from at least three different experiments. For determination of the statistical significance, the Student’s paired two-tailed *t*-test or one-way analysis of variance (ANOVA) followed by Tukey’s honestly significant difference (HSD) test using SPSS 26.0 software (IBM, Chicago, IL, USA) were used. A *p*-value less than 0.05 for all comparisons was considered significant.

## Figures and Tables

**Figure 1 ijms-26-08145-f001:**
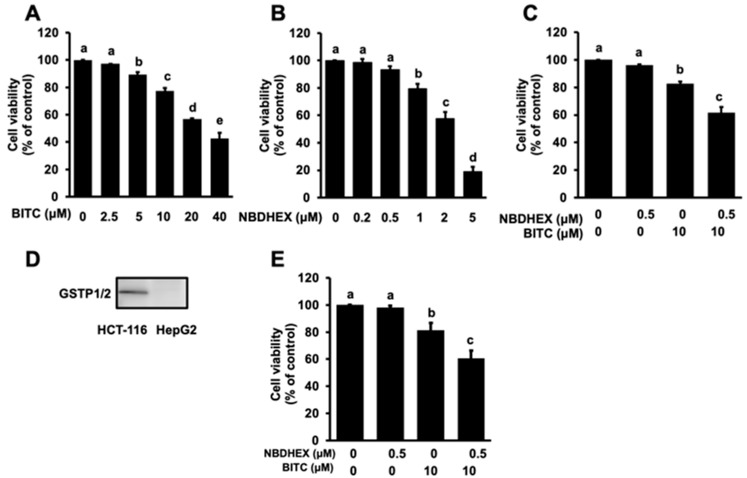
Enhancing effect of NBDHEX on BITC-induced antiproliferation in human colorectal cancer HCT-116 cells and human liver cancer HepG2 cells. (**A**) HCT-116 cells were treated with BITC at the indicated concentrations for 24 h. (**B**) NBDHEX was used to treat the cells for 24 h. (**C**) After pretreatment with NBDHEX (0.5 μM) for 1 h, cells were treated with BITC (10 μM) for 24 h. Cell viability was determined using the MTT assay. (**D**) Basal expression of GSTP1/2 in HCT-116 cells and HepG2 cells. (**E**) After HepG2 cells were pretreated with NBDHEX (0.5 μM) for 1 h, they were treated with BITC (10 μM) for 24 h. Cell viability was determined using the MTT assay. The different letters over the bars correspond to significant differences between treatments for each condition (*p* < 0.05, Tukey’s HSD, *n* = 3).

**Figure 2 ijms-26-08145-f002:**
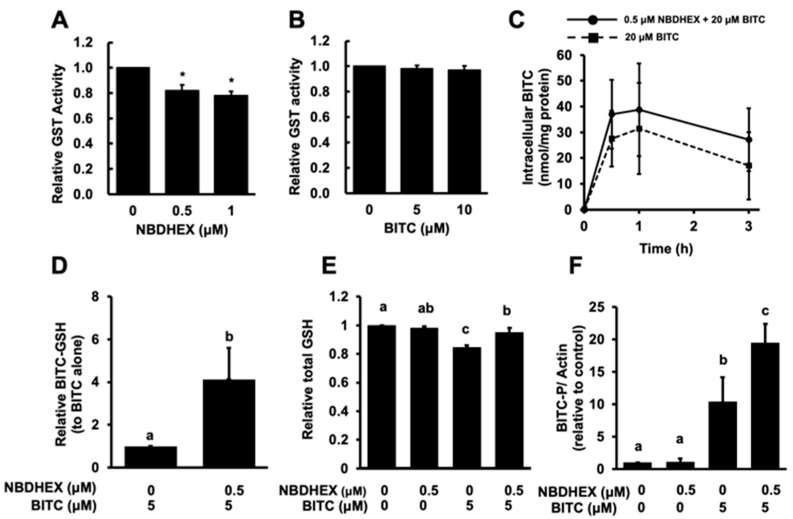
Modulatory effect of NBDHEX on BITC metabolism. Effects of NBDHEX (**A**) and BITC (**B**) on total GST activity. The total GST activity was determined using the CDNB assay. * *p* < 0.05 vs. control, Student’s *t*-test (*n* = 3). (**C**) Effect of NBDHEX on the intracellular BITC level. After pretreatment with NBDHEX (0.5 μM) for 1 h, cells were treated with BITC (20 μM) for 0.5, 1, and 3 h. The intracellular level of BITC was evaluated using the cyclocondensation assay. (**D**) Effect of NBDHEX on the BITC–GSH conjugate level in the cells. After pretreatment with NBDHEX (0.5 μM) for 1 h, cells were treated with BITC (5 μM) for 1 h. The BITC–GSH level was quantified using LC–MS/MS. (**E**) Effect of NBDHEX on the GSH level in the cells. After pretreatment with NBDHEX (0.5 μM) for 1 h, cells were treated with BITC (5 μM) for 1 h. The total GSH level was quantified using the DTNB-glutathione reductase assay. (**F**) Effect of NBDHEX on the level of BITC-modified proteins. After pretreatment with NBDHEX (0.5 μM) for 1 h, cells were treated with BITC (20 μM) for 1 h. The BITC-modified proteins were analyzed by western blotting. The different letters over the bars correspond to significant differences between treatments for each condition (*p* < 0.05, Tukey’s HSD, *n* = 3).

**Figure 3 ijms-26-08145-f003:**
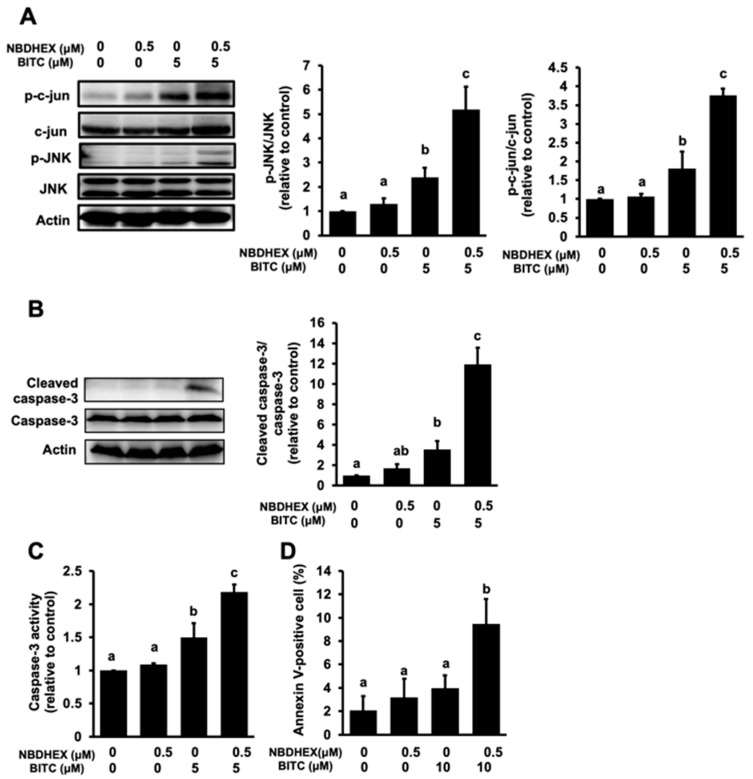
Enhancing effect of NBDHEX on BITC-induced apoptosis and its signaling pathways. (**A**) Enhancement of BITC-induced activation of the JNK pathway by NBDHEX. After pretreatment with NBDHEX (0.5 μM) for 1 h, cells were treated with BITC (5 μM) for 1 h. The phosphorylated and total protein levels of c-jun and JNK as well as actin were analyzed with western blotting. (**B**,**C**) NBDHEX enhanced the BITC-induced caspase-3 activation. After pretreatment with NBDHEX (0.5 μM) for 1 h, cells were treated with BITC (5 μM) for 24 h. Caspase-3, cleaved caspase-3, and actin were analyzed with western blotting (**B**). The caspase-3 activity was assessed with an enzyme assay using a substrate specific for caspase-3 (**C**). (**D**) Effects of NBDHEX and BITC on the induction of apoptosis. After pretreatment with NBDHEX (0.5 μM) for 1 h, cells were treated with BITC (10 μM) for 24 h. The percentage of the cell population that exhibited apoptotic characteristics was estimated using a commercially available apoptosis detection kit (Annexin V-FITC Apoptosis Detection Kit, Nacalai Tesque Inc., Kyoto, Japan) with observations performed using a fluorescence microscope (Biozero BZ-8000, KEYENCE, Osaka, Japan). The different letters over the bars correspond to significant differences between treatments for each condition (*p* < 0.05, Tukey’s HSD, *n* = 3).

**Figure 4 ijms-26-08145-f004:**
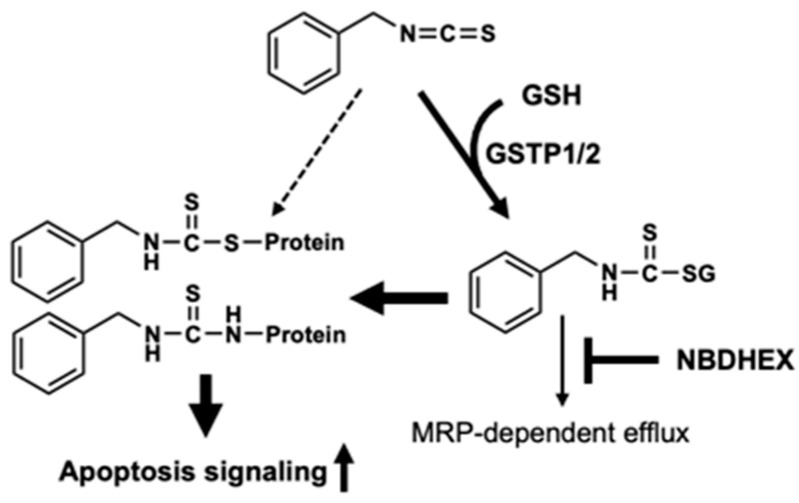
Schematic illustration of the potentiation of BITC-induced apoptosis by NBDHEX and its involvement in the inhibition of the BITC metabolite efflux. The thickness of the arrows varies because some pathway is inhibited by a certain inhibitor, or because another pathway is enhanced as a result. The dash arrows indicate a possible direct flow that has not been proven in this study.

## Data Availability

The original contributions presented in the study are included in the article. Further inquiries can be directed to the corresponding author.

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
