# Peer review of "Augmentation of the Benzyl Isothiocyanate-Induced Antiproliferation by NBDHEX in the HCT-116 Human Colorectal Cancer Cell Line"

_ijms, 2025, doi:10.3390/ijms26178145_

Round 1

Reviewer 1 Report

Comments and Suggestions for Authors

see pdf

Author Response

Reviewer #1

The authors presented a manuscript titled “Augmentation of the Benzyl Isothiocyanate-induced Anti-proliferation by A Pi-class Glutathione S-Transferase Inhibitor in Human Colorectal Cancer Cells”. The study presents data on cell viability, GST activity, GSH levels, BITC-protein adduct formation, and apoptosis signaling (JNK, caspase-3), and concludes that GSTP1 inhibition sensitizes cells to BITC. The data are mostly convincing, and the conclusions are generally supported by the results. While this manuscript may be relevant to those who are interested in the field of either BITC or NBDHEX, however, several issues, particularly in experimental depth and mechanistic interpretation, need to be addressed before publication.

Response: We would like to express our gratitude to the reviewers for their careful reading of the manuscript and constructive comments, especially those regarding the weaknesses in our discussion and how to improve them. Based on the reviewers' comments, we have conducted additional experiments and revised the manuscript. Consequently, unexpected experimental results were obtained. Although the conclusion that NBDHEX enhances the antiproliferation effect of BITC remains unchanged, the results suggest that there is no specificity for colorectal cancer cells and that the role of GSTP1 is negative. Anyway, we believe that this paper has been significantly improved, and we hope that the revisions we have made and the attached responses address the concerns raised.

Overall comments:

1) The study is entirely of study’s proposed mechanism is based on HCT-116 cells and additional cell lines with difference in GSTP1 needs to be added to show robustness of the conclusion. One of the main points that the authors proposed was that GSTP1/2 plays a role to reduce efficacy of BITC-induced apoptosis. Hence, the authors should be including multiple cell lines and even knock down experiments to robustly suggest this is indeed acting through GSTP1/2. As such, also does not fit the title “Human Colorectal Cancer Cells”.

Response: We agree with the reviewer’s idea that, to conclude that GSTP1/2 plays a role in the reduction of BITC’s efficacy in human colorectal cancer cells, we need to perform the experiment using additional cell line with difference in GSTP1. Since the detailed analysis using knock down experiments will require more than several months, an experiment using the human liver cancer cell line HepG2, which expresses GSTP1/2 only below the detection limit (revised Figure 1D), was conducted. As shown in the revised Figure 1E, NBDHEX also significantly enhanced the BITC-induced antiproliferation in HepG2 cells. These results implied that the NBDHEX’s effect involves the mechanism other than inhibition of GSTP1/2. Therefore, the role of GSTP1/2 in reducing the efficacy of BITC might also be uncertain, as is the specificity on colorectal cancer cells. Based on this experimental data, the title, introduction and discussion sections were substantially revised. The potential off-target effect of NBDHEX has also been discussed in the revised manuscript. 

2) Mechanistic clarity in terms of BITC accumulation vs. redistribution: The authors claim that NBDHEX increases free BITC or its cytotoxic targets (BITC-modified proteins). However, Figure 3A shows no significant increase in total BITC, yet Figure 3C suggests enhanced BITC-protein conjugates. Further experiments need to be performed to suggest that NBDHEX likely redistributes BITC metabolism from GST-GSH conjugation toward protein adduction, rather than increasing total BITC. Consider measuring BITC-GSH conjugates vs. BITC-protein adducts directly (e.g., LC-MS/MS or targeted metabolomics) to strengthen this point.

Response: According to the review’s suggestion, we quantified the BITC-GSH conjugate in the cells treated with BITC and NBDHEX by LC-MS/MS, as shown in the revised figure 2D. Contrary to expectations, NBDHEX did not suppress the accumulation of BITC-GSH but rather promoted it. This data suggested that NBDHEX might increase the intracellular accumulation of BITC-GSH by inhibiting its excretion instead of inhibiting GST, and that thiol exchange reactions or free BITC released from BITC-GSH by chemical equilibrium might contribute to protein modification. This point and the corresponding discussion have been revised along with Figure 7.

3) The authors suggest synergistic interaction (i.e. “BITC + NBDHEX treatment is greater than additive”). However, this needs to verified by using formal synergism test – i.e. Bliss independence or Chou-Talalay method to calculate a synergy score. At minimum, provide dose–response curves for both BITC and NBDHEX alone and in combination.

Response: We are focusing purely on the potentiating effect of NBDHEX and do not wish to emphasize synergistic interactions. Therefore, we have revised the manuscripts to tone down these expressions in the results and discussion sections. 

4) Experimental Validation for western blots and other experiments – please include at least 3 replicates for westerns perform including all their blots (including replicates) as per required by MDPI. Some of the blots lack clear densitometric quantification or show low signal-to-noise. Providing replicates will valid this. 

Response: We have confirmed that all blots are in a condition that allows for clear densitometric analysis in accordance with MDPI standards. We would also like to add that all blots shown are unadjusted, without any contrast adjustments.

Other points are mentioned in below per sections comments:

Introduction:

- Overall it is unclear which drug is the main drug? i.e. is the authors trying to suggest BITC or GST as an inhibitor?

Response: BITC is the main compound we are focusing on. Therefore, we have rearranged the order of the paragraphs in the introduction and moved the introduction to BITC to the beginning.

- It is unclear why the authors investigate colorectal cancer. What is the relevance of this?]

Response: As mentioned above, the additional experiment demonstrated that NBDHEX showed the same enhancing effect on BITC in the human liver cancer cell line HepG2, which expresses GSTP1/2 only below the detection limit. Therefore, we have not stuck so much to colorectal cancer in the revised manuscript.

- Is GST inhibitors used clinically? If not, why should readers care about GST inhibitor’s resistance? i.e. is this not used in clinic because of resistance in phase 2 trials?

Response: The answer is “No”. One of the significant  obstacles that GST inhibitors encounter in clinical trials is their insufficient specificity (PMID: 33946704). We have added this information into the discussion section of the revised manuscript.

Methods:

- Overall description of methods is fine, with the exception of 4.3 (expand) , 4.4(explain more on cyclocondensation assay), 4.5 (elaborate measurement, how ?).

Response: According to the suggestion by the reviewer, we have revised the method section.

- Please describe how many technical replicates per experiment and how many times experiment have been performed for all assays.

Response: As suggested, we have added the number of experiment replicates.

- Further experimental improvements are discussed in results – i.e. the experiments that needed to be performed.

Response: As mentioned above, we have performed the additional experiments. Based on the data, we have revised the discussion section as shown in the revised manuscript.

Results:

Section 2.1: Enhancing effects of NBDHEX on the BITC-induced antiproliferation

- Include IC50 including its 95% Confidence interval when describing the single drug dose response.

- Please include the 95% confidence interval when describing the data (i.e. line 97-98) – this will allow readers to know how much the error and magnitude of change is.

- It is important for authors to describe the viability +/- 95% CI of the dose that they used for combination.

- Also describe number of technical and experimental replicates used.

Response: As suggested, we have added IC50 values as well as 95% confidential interval to make the data variability and comparison much clearer. We have also added the number of experiment replicates.

Section 2.2: Modulating effect of NBDHEX on the intracellular metabolism of BITC

- Extra description in methods or here of how the assay works is needed.

It is unclear whether this is normalized to number of cells – i.e. if treatment kills the cells, then one would expect lesser GST.

- Figure 2 and 3 should be combined together as they are related.

- The authors used different doses of BITC in their experiments in Figure 2B and Figure 3. To make appropriate comparison, the authors should perform the same dose. One cannot say that BITC at 20 uM won’t affect GST activity.

- The authors should also perform more comprehensive doses to show that there is a dose dependent relationship between NBDHEX and GST activity.

Response: Thank you very much for your valuable and insightful comments. As suggested, I have combined Figures 2 and 3. I have also provided detailed descriptions of the experiments. For biochemical experiments, I used short-term treatments that do not cause cell death, and I standardized the experiments by unifying the protein concentrations. Regarding the inhibition of GST activity by high-concentration BITC, since this is an in vitro experiment and is not directly related to the main theme of this study, I would like to leave it as it is. I have used the same concentration (5 μM) of BITC for the experiments of the revised Figures 2D, 2E and 2F.

Section 2.3: Modulating effects of NBDHEX on the apoptosis-related pathways and section 2.4: Enhancing effects of NBDHEX on the BITC-induced apoptosis

- These are related sections and should be combined together. (i.e. Figure 4, 5, 6 should be combined, it is related to apoptosis.)

- The authors used different doses in these sections, which will not allow direct comparison of each section. Please perform experiments with the same dose so that we can have proper comparison.

- For figure 6, the authors should perform flow cytometry using caspase3/7 dye to show apoptosis in an accurate quantitative manner.

- Importantly, these data do also show some indication that each compound alone may cause apoptosis (likely at the higher doses) – the authors should confirm this at a higher dose (i.e. 1uM NBDHEX). While it may show synergism at low dose, there may be some off target effect too.

Response: Frist of all, according to the review’s advice, I have combined Figures 4, 5 and 6 into the revised Figure 3. We used BITC at the same concentration for the apoptosis experiment as used in MTT assay. We believe that the effect of NBDHEX was observed even at low concentrations in the analysis of signal transduction molecules due to detection sensitivity issues. At the very least, we standardized the concentrations in Figures 2D, 2E, 2F, 3A, 3B, and 3C. It would take several months to redo all the experiments, so we would like to leave some of the data as is. Thank you for your helpful suggestions regarding flow cytometry. We have already detected caspase-3 biochemically, so we used the current method.

Discussion:

Discussion will be reevaluated once other experiments have been completed or added.

Just some comments for discussion

- Line 186-189 – Physiologically, is it possible to treat a patient at the uM dose (the authors also noted in line 263)? If not, why will these be relevant?

- Line 192 -195 – please expand on what sort of dose and in what context it is toxic?

- Figure 7 is a little misleading since the authors didn’t really show that NBDHEX inhibits GSTP1/2 (i.e. through knockout experiments).

Response: Based on additional experiments and data revisions, we have significantly revised the discussion. I don't think it's physiologically possible to achieve a μM concentration of BITC. Unfortunately, BITC alone cannot exert its anticancer effect unless it is in this concentration range. The reason for this is thought to be the involvement of the MDR mechanism. The purpose of this study is to explore compounds that improve the MDR mechanism and enable BITC to exert its effect at lower concentrations through combination treatment. Finally, Figure 7 has been totally revised and replaced by the revised Figure 4.

Reviewer 2 Report

Comments and Suggestions for Authors

The abstract is informative. The introduction contains enough information to understand the context of the research.

In the results section, all figure captions need improvement. More informative text is needed. Information on how many experiments were performed and how many times they were repeated. Information on the statistical significance of the letters over the bars.

Figures 2 and 3 are complementary as figure 4 and 5. Is it possible to join them?

Have you analyzed reduced GSH? It would be an interesting piece of information to add to the text.

The methodology is not well described. A better description is needed in order to be able to reproduce it. Even the items that are cited in the articles provide a very brief description of the methods. In the methodology item, it is important to include the clones of the antibodies used in order to have reproducibility

Comments on the Quality of English Language

The text itself is not badly written, but the subtitles need improvement.

Author Response

Reviewer #2

In the results section, all figure captions need improvement. More informative text is needed. Information on how many experiments were performed and how many times they were repeated. Information on the statistical significance of the letters over the bars.

Response: Thank you very much for your valuable and insightful comments. As suggested, we have added the number of experiment replicates.

Figures 2 and 3 are complementary as figure 4 and 5. Is it possible to join them?

Response: As suggested, I have combined Figures 2 and 3 into Figure 2. Also, Figures 4, 5 and 6 have been combined into Figure 3.

Have you analyzed reduced GSH? It would be an interesting piece of information to add to the text.

Response: No, we just checked the total GSH (GSH+GSSG). We have previously shown that 5μM BITC decreases the reduced GSH level, possibly through the formation and efflux of its conjugate, but does not affect the GSSG levels (PMID: 22296293). Therefore, we did not analyze this in the present study.

The methodology is not well described. A better description is needed in order to be able to reproduce it. Even the items that are cited in the articles provide a very brief description of the methods. In the methodology item, it is important to include the clones of the antibodies used in order to have reproducibility.

Response: According to the suggestion by the reviewer, we have revised the method section to add brief experimental conditions as well as antibody product numbers. 

Round 2

Reviewer 1 Report

Comments and Suggestions for Authors

See pdf 

Author Response

The authors only included an additional liver cancer cell line which does not have expression of GSTP1. The current title reads “Augmentation of the Benzyl Isothiocyanate-induced Antiproliferation by NBDHEX in Human Cancer Cells”, and to be more specific the authors should change it to “Augmentation of the Benzyl Isothiocyanate-induced 2 Antiproliferation by NBDHEX in HCT-116 colorectal cancer cell line”. This is because subsequent experiments attempting to describe mechanism are only performed on one cell line, which again, is not considered robust. Ideally, one should at least also include normal cells and at least 2 cell lines (with GST expression) with GST expression to prove the statements made. This flaw should be recorded as limitation in the discussion.

Response: Thank you very much for your review and constructive comments. We have sincerely revised the manuscript based on the reviewers' comments. We fully agree with the reviewers' idea that the experiments attempting to describe the mechanism were performed using only a single cell line, which is considered unreliable. Accordingly, the title has been replaced by “Augmentation of the Benzyl Isothiocyanate-induced Antiproliferation by NBDHEX in HCT-116 Human Colorectal Cancer Cell Line”. In addition, this point has been added as another limitation of this study in the discussion section of the revised manuscript. 

Indeed, this figure have been added. A few questions about the Figure 2D:

– Can the authors explain how relative BITC-GSH (to BITC alone) this is calculated? If this is a relative value of BITC-GSH divide by BITC, any values < 1 means that BITC > BITC-GSH.

- Rather than presenting in relative quantity, the authors should present this in absolute quantity. i.e. pmol/mg protein or nmol/10⁶ cells for Absolute BITC–GSH quantity and absolute intracellular BITC levels. If it is indeed relative, would the first two values be infinite (i.e. 0/0)?

Response: Relative BITC-GSH is expressed based on the BITC-GSH amount detected in the BITC-only treatment group being set at 1. To avoid misunderstanding by readers, the data for the BITC-untreated groups has been removed from Figure 2D. We also believe that it is better to express the amount of BITC-GSH in absolute values. However, there is no commercially available standard product necessary for identification. Although we have synthesized our own, we have not been able to measure its exact weight. Therefore, we are unable to perform accurate quantification and cannot indicate absolute values.

The authors have uploaded the western blots, and there seems to be quite a stark difference in quality of the blot. Interestingly, I would just like to highlight that some of the blots have ladders, and some don’t. Some blots are perfect across replicates – i.e. Actin (Fig 4), Caspase 3 (Fig 5). But some are overexposed (Cleaved caspase 3 (fig 5), Actin (fig 4), p-JNK (fig 4).), or didn’t have ladder (Actin Fig 3, p-c-jun Fig 4,c-jun Fig 4). I’ll leave this to the discretion of the editor.

Response: Thank you for carefully examining the original Western blot analysis photos. To clearly show the expression of the target protein, pictures with long exposure times are shown, resulting in the detection of non-specific positive signals. However, since ladder-like non-specific signals do not affect the quantification of the target protein, we believe that this is not a problem.

Answered – The authors should also state that they have ““unifying the protein concentrations” in the methods. And be specific on how this is done. i.e. measured total protein in each sample (e.g., by BCA or Bradford assay) and then normalized the assay readouts to the same total protein amount before running the GST activity or BITC–GSH quantification.?

Response: Thank you very much for pointing out the inadequacies in our description. In accordance with your comments, we have described the protein quantification method and its application to each assay.

I think this statement in the author’s response “The purpose of this study is to explore compounds that improve the MDR mechanism and enable BITC to exert its effect at lower concentrations through combination treatment.”, is useful to include in the manuscript to really highlight the purpose of this study.

Response: Thank you very much for your thoughtful suggestion. We have included this in the introduction to highlight the purpose of this study.

Additional comments based on edited manuscript:

- Line 130- 131 : “We then examined the effect of NBDHEX on the intracellular GSH level, because GSH is a substrate for GST and thus its intracellular level is reduced by the BITC conjugation”

Is there an error in this sentence ? – particularly “because GSH is a substrate for GST”.

Response: We have deleted this point in the revised manuscript.

- Line 132 – 133: is this statement and reference to figure updated?

- There is no reference to figure 2F in results section.

Response: Thank you very much for indicating our mistakes. Figure 3B is incorrect, and the correct description is Figure 2E. Thus, we have replaced “Figure 3B” by “Figure 2E”. In addition, we forgot to refer to Figure 2F. We have corrected this in the revised draft.

Round 3

Reviewer 1 Report

Comments and Suggestions for Authors

No further comments